# Fault Identification for a Closed-Loop Control System Based on an Improved Deep Neural Network

**DOI:** 10.3390/s19092131

**Published:** 2019-05-08

**Authors:** Bowen Sun, Jiongqi Wang, Zhangming He, Haiyin Zhou, Fengshou Gu

**Affiliations:** 1College of Liberal Arts and Sciences, National University of Defense Technology, Changsha 410073, China; sunbowen2017@sina.com (B.S.); hzmnudt@sina.com (Z.H.); gfkd_zhy@sina.com (H.Z.); 2Beijing Institute of Spacecraft System Engineering, China Academy of Space Technology, Beijing 100094, China; 3School of Computing and Engineering, University of Huddersfield, West Yorkshire HD1 3DH, UK; F.Gu@hud.ac.uk

**Keywords:** closed-loop control system, fault diagnosis, deep neural network, sliding window, identification performance

## Abstract

Fault identification for closed-loop control systems is a future trend in the field of fault diagnosis. Due to the inherent feedback adjustment mechanism, a closed-loop control system is generally very robust to external disturbances and internal noises. Closed-loop control systems often encourage faults to propagate inside the systems, which may lead to the consequence that faults amplitude becomes smaller and fault characteristics difference becomes more inapparent. Hence, it has been challenging to achieve fault identification for such systems. Traditional fault identification methods are not particularly designed for closed-loop control systems and thus cannot be applied directly. In this work, a new fault identification method is proposed, which is based on the deep neural network for closed-loop control systems. Firstly, the fault propagation mechanism in closed-loop control systems is theoretically derived, and the influence of fault propagation on system variables is analyzed. Then deep neural network is applied to find fault characteristics difference between different data modes, and a sliding window is used to amplify the fault-to-noise ratio and characteristics difference, with an aim to increase the identification performance. To verify this method, the simulations that are based on a numerical simulation model, the Tennessee industrial system and the satellite attitude control system are conducted. The results show that the proposed method is more feasible and more effective in fault identification for closed-loop control systems compared with traditional data-driven identification methods, including distance-based and angle-based identification methods.

## 1. Introduction

With the development of science and technology, the complexity of the industrial systems has been increasing rapidly. The fault of these complex systems can lead to the decline in product quality and may cause significant property damage or casualties. Therefore, it has become a hot research topic to improve safety and reliability of system operations and to detect or even to identify faults accurately in a timely manner [1,2,3,4].

In order to achieve the predetermined production goals and meet the stability and the robustness for the industrial systems, the closed-loop control is generally applied [5,6]. Through the closed-loop control, the influence of external disturbances and internal noises on the operation of the system is reduced, which makes the system much more robust. At present, a large number of closed control loops have been widely used in the industrial production processes, ranging from mechanical machine control to spacecraft control [7,8].

Due to the nature of closed control loops, the performance of fault diagnosis is degraded. The main reasons are as follows [9]:

(1) The closed control loop usually makes the system more robust to external disturbances and internal noises. When the fault happens in the early stage or the fault amplitude is small, the fault signal will be covered by external control signals, which is difficult for the fault to be detected, resulting in lower fault detection rates;

(2) Due to the inherent feedback adjustment mechanism, a closed-loop control system encourages faults to propagate inside the system, leading to the faults existing in many variables/signals and the faults amplitude becoming smaller, or resulting in the faults characteristics difference being much inapparent and the faults coupling with each other. Such feedback adjustment ability adds to the difficulty for fault identification.

This paper focuses on the second issue in closed-loop control systems. The main aim is to achieve fault identification in closed-loop control systems and improve fault identification performance.

For the structural health monitoring in opto-mechanical systems, the system scanning technology is based on a mechanical rotary mirror and the computational processing of an optoelectronic signal is applied [10]. The 3D measurement is improved with the multivariate outlier mining and regression feedback. New research performed in the signal and data processing of the 3D measurement system demonstrated the effectiveness of signal processing strategies as a tool for a feedback loop in any measurement system.

For position detection in real life application for structural health monitoring (SHM) by a novel method, support vector machine (SVM) regression was applied to predict measurements errors for accuracy enhancement in optical scanning systems [11]. The accuracy can be enhanced based on the power spectrum centroid calculation. The SVM regression method is proved that it can be used to increase measurement accuracy for optical scanning systems.

There are a lot of research on fault identification, and the core problem of fault identification lies in the extraction of fault characteristics.

Literature [12] proposed a fault identification method based on kernel-independent component analysis (KICA). This method extracts the direction information of the faults as the identification characteristics, and then uses the training data to build a model for fault identification. However, as the complexity of closed-loop control systems increases, it becomes more and more difficult to build an accurate model only based on sensor data.

With the development of machine learning research, more and more scholars apply it to the field of fault diagnosis. Literature [13] was inspired by contribution analysis (CA) method. This method decomposed the distance of k-nearest neighbor (kNN) and used it as the fault characteristics for fault identification. It addressed the shortcomings of traditional CA method in fault identification. However, the method based on the distance judgment could not properly identify the coupled fault modes. A closed-loop control system has high complexity and multiple coupled modes. Hence the above method is not suitable for fault identification in closed-loop control systems.

In non-homogeneous discrete systems, Literature [14] detected and identified faults in the case of a large number of sensors disturbances. The residual of the observed variable was calculated by estimating the residual characteristics of the state variable, and an appropriate threshold [7] was selected to compare the difference between the residual and the threshold in each dimension.

Literature [15] approached the fault detection and identification (FDI) problem for nonlinear systems by using neural networks. The proposed FDI approach employed a recurrent neural network-based observer for detecting, isolating and identifying actuator faults in presence of disturbances and uncertainties in model and sensor measurements. The method used a modified dynamic backpropagation scheme to update the weights of the neural network, and identified the fault characteristics by detecting the updated parameter characteristics of the neural network. However, for closed-loop control systems, different modes may have strong coupling, resulting in parameter changes not meeting the detection and identification requirements.

According to the statement above, most of existing fault identification methods for open-loop systems cannot be directly applied to closed-loop control systems. For closed-loop control systems, the dynamicity of the system poses great challenges in constructing an accurate physical model and determining the model parameters, and the mathematical model established by data fitting may not be malleable or expandable for test data, which leads to the consequence that state vectors or system parameters cannot be estimated accurately. In addition, a closed-loop control system is generally robust to the external disturbances [16], so that the fault amplitude gradually becomes smaller during fault propagating inside the system; moreover, the fault characteristics difference between different modes can be reduced, and the fault modes could be coupled with each other. Those all add to the difficulty of fault identification for closed-loop control systems.

Based on these observations, in this paper, a fault identification method based on a deep neural network is proposed. The deep neural network [17] can learn independently, and can obtain the characteristics of each data or fault mode through repeated training. It is expected to obtain a better fault identification performance.

The traditional deep neural network method only trains each set of data. If it is directly applied to a dynamic closed-loop system, the dynamics between each group of data will be neglected [18,19]. In addition, the method is proposed to amplify the fault-to-noise ratio and the characteristics difference through applying a sliding window. This paper proposes techniques to improve the deep neural network when the fault magnitude is small and the fault characteristics difference is inapparent in close-loop control systems. The method proposed in this paper has a better identification effect on tagged data. The tagged training data is used for studying with the supervision for each mode, and then the test data is fed into the trained deep neural network for fault mode classification, and then fault identification for closed-loop control system can be realized.

The main contributions of this paper are as follows: Firstly, the propagation mechanism of sensor failure and process fault in closed-loop control systems, and their influence on system output variables are analyzed theoretically. Secondly, a deep neural network fault identification method based on sliding window is proposed, and the advantage of this method for identifying strong coupled modes is demonstrated. Thirdly, the improved deep neural network method is applied to the numerical simulation system, the Tennessee industrial system and the satellite attitude control system to verify fault identification in closed-loop control systems. Finally, the paper also makes contributions for the problem of how to process the data from sensors.

The remainder of this paper is organized as follows: in Section 2, the propagation mechanism of faults and their influence on output variables in close-loop control systems are analyzed theoretically. In Section 3, traditional distance-based and angle-based fault identification methods are introduced. In Section 4, by introducing the deep neural network structure, a deep neural network fault identification method based on sliding window is proposed. Section 5 presents experiments on the numerical simulation model systems, the Tennessee industrial systems and the satellite attitude control systems, and then results are reported and discussed. Finally, the conclusion is drawn in Section 6.

## 2. Closed-Loop Control System and Fault Propagation

### 2.1. Closed-Loop Control System Model

Firstly, a discrete linear model for a control system is established as follows [20,21]:(1)$$\left\{\begin{array}{c} x_{k+1}=Ax_{k}+Bu_{k}+f_{k}^{x}+w_{k}\\ y_{k}=Cx_{k}+f_{k}^{y}+v_{k} \end{array}\right.,$$
where $x_{k}\in R^{n_{x}}$ is the state vector, $u_{k}\in R^{n_{u}}$ is the control input vector, $y_{k}\in R^{n_{y}}$ is the measurement output vector, $w_{k}\in R^{n_{x}}$ and $v_{k}\in R^{n_{y}}$ are the independent process noise and the measurement noise with zero mean, and the covariance matrices are the normal distribution of $Q\in R^{n_{x}\times n_{x}}$ and $R\in R^{n_{y}\times n_{y}}$ respectively; $f_{k}^{x}\in R^{n_{x}}$ and $f_{k}^{y}\in R^{n_{y}}$ are the process fault and the sensor fault respectively (When the system operates normally, i.e., $f_{k}^{x}=0$ and $f_{k}^{y}=0$), A,B,C are the parameter matrices with appropriate dimensions, which are all unknown.

The basic principle of a closed-loop control system is that the system constructs a deviation vector between the measurement output vector and its expected output vector, uses the deviation feedback on the control input vector, and then introduces a closed-loop control rate to reduce the influence from external interference.

When adding the closed control loop into model (1), the closed-loop control system equation can be written as follows:(2a)$$x_{k+1}=Ax_{k}+Bu_{k+1}+f_{k}^{x}+w_{k+1}$$
(2b)$$y_{k}=Cx_{k}+f_{k}^{y}+v_{k}$$
(2c)$$x_{k+1}^{c}=A^{c}x_{k}^{c}+B^{c}y_{k}$$
(2d)$$u_{k+1}={\tilde{u}}_{k+1}+C^{c}\left({\tilde{x}}_{k+1}^{c}-x_{k+1}^{c}\right)+D^{c}\left({\tilde{y}}_{k}-y_{k}\right),$$
where $x_{k}^{c}\in R^{n_{x^{c}}}$ is the state control vector, ${\tilde{u}}_{k+1}\in R^{n_{u}}$ is the expected control input vector, ${\tilde{x}}_{k}^{c}\in R^{n_{x^{c}}}$ is the expected state control vector, ${\tilde{y}}_{k}\in R^{n_{y}}$ is the expected measurement output vector, $A^{c},B^{c},C^{c},D^{c}$ are the parameter matrices with appropriate dimensions. As can be seen from Model (2), the closed-loop control system feeds back the deviation vector (i.e., ${\tilde{y}}_{k}-y_{k}$) to the control input vector $u_{k+1}$ (i.e., the control input vector at the next k+1-th moment), thereby adjusting the operation situation of the closed-loop control system.

### 2.2. Propagation Mechanism of Sensor Fault

Suppose that the symbol $\hat{\cdot }$ represents the corresponding vector when the closed-loop control system operates normally. Consider that a sensor fault occurs at the *k*-th moment (i.e., $f_{k}^{y}\neq 0$), then the measurement output vector $y_{k}$ can be represented as(3)$$y_{k}=Cx_{k}+f_{k}^{y}+v_{k}={\hat{y}}_{k}+f_{k}^{y}.$$

According to Equation (2c), the state control vector $x_{k+1}^{c}$ is affected by the measurement output vector as follows:(4)$$x_{k+1}^{c}=A^{c}x_{k}^{c}+B^{c}{\hat{y}}_{k}+B^{c}f_{k}^{y}={\hat{x}}_{k+1}^{c}+B^{c}f_{k}^{y}.$$

According to Equation (2d), the control input vector $u_{k+1}$ is affected by both the measurement output vector and the state control vector as follows:(5)$$u_{k+1}={\tilde{u}}_{k+1}+C^{c}\left({\tilde{x}}_{k+1}^{c}-x_{k+1}^{c}\right)+D^{c}\left({\tilde{y}}_{k}-y_{k}\right)={\hat{u}}_{k+1}-\left(C^{c}B^{c}+D^{c}\right)f_{k}^{y}.$$

Then, the state vector at the k+1-th moment (i.e., $x_{k+1}$) can be obtained as follows:(6)$$x_{k+1}=Ax_{k}+Bu_{k+1}+w_{k+1}={\hat{x}}_{k+1}-B\left(C^{c}B^{c}+D^{c}\right)f_{k}^{y}.$$

According to the influence relationship above, the sensor fault at *k* moment (i.e., $f_{k}^{y}$) will ultimately affect the measurement output vector at the k+1-th moment (i.e., $y_{k+1}$) as follows:(7)$$y_{k+1}=Cx_{k+1}+v_{k+1}={\hat{y}}_{k+1}-\mathrm{CB}\left(C^{c}B^{c}+D^{c}\right)f_{k}^{y}.$$

Let(8)$$H=\mathrm{CB}\left(C^{c}B^{c}+D^{c}\right).$$

Make the eigenvalue decomposition (ED) on the matrix $H\in R^{n_{y}\times n_{y}}$, s.t.(9)$$H=Q_{H}\Lambda_{H}Q_{H}^{-1},$$
where $Q_{H}$ is an eigenvector matrix, $\Lambda_{H}=\left[\begin{array}{cc} \Sigma & 0\\ 0 & 0 \end{array}\right]\in R^{n_{y}\times n_{y}}$, $\Sigma =diag(\lambda_{1},\lambda_{2},\cdots ,\lambda_{l})$, $\lambda_{i}\;\left(i=1,2,\cdots ,l\right)$ is the eigenvalues of H and $l=rank\left(H\right)$.

**Remark** **1.***From the propagation process of sensor fault in a closed-loop control system above, if the sensor fault occurs at the k-th moment (i.e., $f_{k}^{y}\neq 0$), the influence of this sensor fault on the measurement output vector at the k+1-th moment (i.e., $y_{k+1}$) is $-Q_{H}\Lambda_{H}Q_{H}^{-1}f_{k}^{y}$. If the closed-loop control system is selected properly, the eigenvalues $\lambda_{1},\lambda_{2},\cdots ,\lambda_{l}$ of H are all less than 1, and then the fault amplitude will be reduced at the k+1-th moment*.

In addition, if this sensor fault $f_{k}^{y}$ propagates until the k+pth moment, the sensor fault amplitude occurring at the *k*-th moment will reduced to ${\left(-1\right)}^{p}Q_{H}\Lambda_{H}^{p}Q_{H}^{-1}f_{k}^{y}$. Since(10)$$\lim_{p\to \infty }\Lambda_{H}^{p}=0,$$
the closed-loop control system can gradually reduce the impact of sensor fault on the system due to the closed control loop.

**Remark** **2.***Consider that the sensor fault occurs both at the k-th moment and the k+1-th moment, and the fault amplitudes are $f_{k}^{y}$ and $f_{k+1}^{y}$, respectively. From the propagation process of the sensor fault in the closed-loop control system above, the influence of the sensor fault with these two moments on the measurement output vector at the k+1-th moment is $f_{k+1}^{y}-\mathrm{CB}\left(C^{c}B^{c}+D^{c}\right)f_{k}^{y}=f_{k+1}^{y}-{\mathrm{Hf}}_{k}^{y}$. If the closed-loop control system is selected properly to make $f_{k+1}^{y}-{\mathrm{Hf}}_{k}^{y}<f_{k+1}^{y}$, and then the influence of the sensor fault on the closed-loop control system will also be reduced at the k+1-th moment*.

### 2.3. Propagation Mechanism of Process Fault

Similar to Section 2.2, the process fault amplitude will decrease because of the closed control loop. The detail process is as Appendix A.

**Remark** **3.***From the propagation process of the process fault in the closed-loop control system above, if the process fault occurs at the kth moment (i.e., $f_{k}^{x}\neq 0$), the influence of this process fault on the measurement output vector at the k+1-th moment (i.e., $y_{k+1}$) is ${\mathrm{Cf}}_{k}^{x}$ and $C\left(A-B\left(C^{c}B^{c}+D^{c}\right)C\right)f_{k}^{x}$ in the k+2-th moment. Based on Equations (Equation 26) and (Equation 30), if the closed-loop control system is selected properly, the eigenvalues $\lambda_{1},\lambda_{2},\cdots ,\lambda_{t}$ of G are all less than 1, and then the impact of the process fault on the measurement output vector will decrease gradually*.

In addition, if this process fault $f_{k}^{x}$ propagates until to the k+p-th moment, the process fault amplitude occurring at the *k*-th moment will be reduced to $Q_{G}\Lambda_{G}^{p-1}Q_{G}^{-1}f_{k}^{x}$. Since(11)$$\lim_{p\to \infty }\Lambda_{G}^{p-1}=0,$$
the closed-loop control system can gradually reduce the impact of the process fault on the system.

**Remark** **4.***Consider that the sensor fault occurs both at the k-th moment and the k+1-th moment, and the fault amplitudes are $f_{k}^{y}$ and $f_{k+1}^{y}$, respectively. The influence of the process fault with both the k-th moment and the k+1-th moment on the measurement output vector at the k+2-th moment is ${\mathrm{Cf}}_{k+1}^{x}+C\left(A-B\left(C^{c}B^{c}+D^{c}\right)C\right)f_{k}^{x}$. If the closed-loop control system is selected properly to make ${\mathrm{Cf}}_{k+1}^{x}+C\left(A-B\left(C^{c}B^{c}+D^{c}\right)C\right)f_{k}^{x}<{\mathrm{Cf}}_{k+1}^{x}$, and then the influence of the process fault on the closed-loop control system will also be reduced*.

**Remark** **5.***The influence of a sensor fault or a process fault on the system is the change of the measurement output vector. In a closed-loop control system, the influence of the two fault modes (sensor fault and process fault) on the system are similar. For example, the sensor fault propagates inside the system with the fault amplitude ${\left(-\mathrm{CB}\left(C^{c}B^{c}+D^{c}\right)\right)}^{p}f_{k}^{y}$, and the process fault propagates inside the system with the fault amplitude $C{\left(A-B\left(C^{c}B^{c}+D^{c}\right)C\right)}^{p-1}f_{k}^{x}$. The fault characteristics difference of these two fault modes is inapparent; therefore, the two types of faults are difficult to distinguish, which increases the difficulty of fault identification*.

## 3. Fault Identification Based on Distance and Angle

### 3.1. Distance Identification Method

Data from different modes tend to differ from each other, and fault identification can be performed by distance characteristics. By calculating the distance between the test data and the data from another mode, the test data can be identified to the closest mode.

Data standardization is necessary to make data nondimensionalization both for the test data and the training data. Firstly, calculate the mean and the covariance for the data with the *k*th mode respectively as follows:(12)$$\left\{\begin{array}{c} {\bar{A}}_{k}=\frac{1}{n_{k}}A_{k}\\ S_{k}=\frac{1}{n_{k}-1}{\left(A_{k}-{\bar{A}}_{k}\right)}^{T}\left(A_{k}-{\bar{A}}_{k}\right) \end{array}\right.,$$
where the subscript indicates the data from the *k*th mode, and the sample number is $n_{k}$ in this mode. $A_{k}$ indicates the training data of this mode, ${\bar{A}}_{k}$ and $S_{k}$ are the mean and covariance of $A_{k}$, respectively. Since $S_{k}$ is a real symmetric matrix, its eigenvalue decomposition can be represented as follows:(13)$$S_{k}=V_{k}\Lambda_{k}V_{k}^{T},$$
where $\Lambda_{k}$ is a diagonal matrix, and $V_{k}$ is an orthogonal matrix. Assume that the test data is x, and let(14)$$z_{k}=\Lambda_{k}^{-1/2}V_{k}^{T}\left(x-{\bar{A}}_{k}\right).$$

Then the Hotelling’s $T^{2}$ statistic can be obtained as follows:(15)$$T_{k}^{2}=z_{k}^{T}z_{k}.$$

This $T^{2}$ statistic obeys the *F*-distribution, and then its threshold can be represented as follows(16)$$T_{\alpha ,k}^{2}=\frac{m\left(n_{k}-1\right)\left(n_{k}+1\right)}{n_{k}\left(n_{k}-m\right)}F_{\alpha }\left(m,n_{k}-m\right)$$
where *m* is the dimension of the training data, α is the confidence level, and $F_{\alpha }\left(m,n_{k}-m\right)$ is the corresponding quantile for *F*-distribution. Thus, the following criterion can be proposed to realize fault identification.

Criterion 1: if $T_{k}^{2}\leq T_{\alpha ,k}^{2}$, then this test data belongs to the *k*th mode; Otherwise, this test data does not belong to the *k*th mode.

### 3.2. Angle Identification Method

The distance identification mentioned in Section 3.1 usually works in the occasions that all modes separate from each other. It is not applicable for the data with some trend characteristics [12]. In this case, it can be considered to use angle characteristics for fault identification [22].

As it is shown in Figure 1, $r_{i}$ and $r_{j}$ represent the direction of the *i*th mode and the *j*th mode, respectively, and r is the direction of the test data, $\tilde{r}=r-r_{i}$.

Let $\theta \left(r_{i},r_{j}\right)$ be the angle between $r_{i}$ and $r_{j}$, and calculate as follows:(17)$$\theta \left(r_{i},r_{j}\right)=\arccos\left(r_{i}^{T}r_{j}/\left(∥r_{i}∥∥r_{j}∥\right)\right)$$

It is obvious that the larger $\theta \left(r_{i},r_{j}\right)$ is, the fault is easier to be identified. For any direction of the test data r, the angle between the test data and the *k*th mode is calculated as follows:(18)$$\theta \left(r,r_{k}\right)=\arccos\left(r^{T}r_{k}/\left(∥r∥∥r_{k}∥\right)\right).$$

Thus, the following criterion can be proposed to realize fault identification.

Criterion 2: Let(19)$$k_{0}=\underset{k=1,\cdots ,n_{k}}{\arg\min}\left\{\theta \left(r,r_{k}\right)\right\}.$$
and this test data belongs to the $k_{0}$-th mode.

**Remark** **6.***From the distance identification and angle identification mentioned in this section, it can be seen that if some modes are coupled with each other, one test data may belong to multiple modes simultaneously. Therefore, the methods based on the distance or angle judgment could not properly identify the coupled fault modes from closed-loop control systems*.

## 4. Deep Neural Network-Based Fault Identification for Closed-Loop Control System

### 4.1. Deep Neural Network Principle

Neural network is a branch of machine learning, and it has been widely used in various fields. However, the traditional single-layer neural network has a limitation of not meeting the classification requirements of complex systems. To solve this problem, deep neural network was proposed in literature [23].

Deep neural network is a multi-layer neural network with two or more hidden layers. Compared with the traditional single-layer neural network, it has the ability of enhancing the fitting effect to make classification and to identify complex data. Based on this, it is possible to train and capture data characteristics of each mode better, and to increase the identification rate of different modes.

A deep neural network is determined by data training with certain learning rules, and then the test data is input through the entire neural network to obtain the output vector [24]. The learning rules are the key in the network establishment process. Here the learning rules are introduced briefly firstly. Deep neural network is a multi-layer perceptron with multiple hidden layers. The network structure is shown in Figure 2.

Where $a={\left[a_{1},a_{2},\cdots ,a_{n}\right]}^{T}$ is the input layer of the deep neural network, *n* is the number of nodes in the input layer, $b={\left[b_{1},b_{2},\cdots ,b_{l}\right]}^{T}$ is the output layer of the deep neural network, *l* is the number of nodes in the output layer, in which the input layer and the output layer are different from that in Equation (Equation 1). Here, the input layer is the control input vector and the measurement output vector, and the output layer is the preset tag. In addition, $W_{i},\;i=1,2,\cdots ,q$ is the weight matrix, $\theta_{i},\;i=1,2,\cdots ,q$ is the bias terms vector, and $\varphi_{i},\;i=1,2,\cdots ,q$ is the activation function and *q* is the number of the hidden layers. Each hidden layer computes the activations on condition that the previous layer has been computed. Denote the input vector from the *j*th hidden layer (which is also the output vector of the j−1th hidden layer) as $v_{j-1}$, and then the output vector of the *j*th hidden layer can be computed as follows:(20)$$v_{j}=\varphi_{j}\left(W_{j}v_{j-1}+\theta_{j}\right),$$
where $W_{j}$ and $\theta_{j}$ are the weight matrix and bias vector of the *j*th layer, and $\varphi_{j}\left(\cdot \right)$ is the predefined nonlinear activation function. Some functions used often in the literature include the sigmoid function $\varphi \left(v\right)=1/\left(1+\exp\left(-v\right)\right)$, the rectifier function $\varphi \left(v\right)=\max\left(0,v\right)$, and the tanh function $\varphi \left(v\right)=\left(\exp\left(2v\right)-1\right)/\left(\exp\left(2v\right)+1\right)$, etc.

In the mode identification region, the softmax activation function is usually used in the last layer of a deep neural network, and its function is as follows:(21)$$\varphi \left(v_{i}\right)=\frac{e^{v_{i}}}{\sum_{k=1}^{M}e^{v_{k}}},$$
where $v_{i}$ is the weighted sum of the *i*th output node, *M* is the number of output nodes. Equation (Equation 21) shows a perfect property that the sum of the outputs is 1, i.e.,(22)$$\varphi \left(v_{1}\right)+\varphi \left(v_{2}\right)+\varphi \left(v_{3}\right)+\cdots +\varphi \left(v_{M}\right)=1,$$
where $0\leq \varphi \left(v_{k}\right)\leq 1,\;k=1,2,\cdots ,M$, and $\varphi \left(v_{i}\right)$ can be regarded as a probability for identifying the mode with the maximum probability.

In the training process, the weight matrix W and the bias vector θ can be updated by using of the gradient descent method, as shown in Equation (Equation 23):(23)$$\left\{\begin{array}{c} E=\frac{1}{m}\sum_{t=1}^{m}E\left(t\right)\\ W_{ij}^{\left(k+1\right)}=W_{ij}^{\left(k\right)}-\alpha \frac{\partial E}{\partial W_{ij}^{\left(k\right)}}\\ \theta_{i}^{\left(k+1\right)}=\theta_{i}^{\left(k\right)}-\alpha \frac{\partial E}{\partial \theta_{i}^{\left(k\right)}} \end{array}\right.,$$
where α is the leaning efficiency, *k* is the iteration number, *m* is the number of training samples, $E\left(t\right)$ is the training error of the *t*th training sample, and E is the mean value of $E\left(t\right)$. The training error is determined by the distance between the theoretical tag β and the actual output b of the deep neural network as follows:(24)$$E\left(t\right)={∥b\left(t\right)-\beta \left(t\right)∥}_{2}$$

If the training error is less than the preset value or the total iteration number reaches the preset number, the training process ends and the network is established.

### 4.2. Improved Deep Neural Network

At present, deep neural network is widely used in many areas, such as image identification, face identification, etc. Sometimes it can be more accurate than that of artificial identification. However, most of the successful cases are aimed at static data, while data from a closed-loop control system is usually dynamic, that is, the current data is affected by previous data or by a period of time. It is difficult to extract the system characteristics only by a single sample data accurately, which is one of the limitations of the traditional deep neural network.

In this paper, for dynamic data from a closed-loop control system, a sliding window can be set to train data with several moments together. The size of the sliding window can be determined empirically or be judged by the correlation of the training data.

Due to the influence of the closed control loop, external disturbance is weakened. Therefore, it is hard to determine whether the fault has occurred from a single data, and even harder for fault identification. An advantage for setting the sliding window is that the characteristics difference between different modes can be accumulated and increased, and then the fault identification rate can be improved.

Take two fault modes as an example, and assume that $y_{k}^{i}={\overset{⌢}{\;y}}_{k}^{i}+e_{k}^{i}$ is the measurement output vector of the *i*th mode and $y_{k}^{j}={\overset{⌢}{\;y}}_{k}^{j}+e_{k}^{j}$ is the measurement output vector of the *j*th mode in the *k*th moment, where ${\overset{⌢}{\;y}}_{k}^{i}$ and ${\overset{⌢}{\;y}}_{k}^{j}$ are the output vectors without noise of the *i*th mode and the *j*th mode, respectively. Then let $r_{k}=y_{k}^{i}-y_{k}^{j}$, $z_{k}={\overset{⌢}{\;y}}_{k}^{i}-{\overset{⌢}{\;y}}_{k}^{j}$, and $e_{k}=r_{k}-z_{k}$. Assume that the $e_{k}$ obeys normal distribution(25)$$e_{k}∼N\left(0,\Sigma \right),$$
where $\Sigma_{k}$ is noise covariance. Then $r_{k}$ obeys the distribution as follows:(26)$$r_{k}∼N\left(z_{k},\Sigma \right).$$

Since the ratio (i.e., $\left|z_{k}/\sqrt{\Sigma }\right|$) between the characteristics difference and the noise is small, it is easily overwhelmed by the noise. This means it is difficult to identify the fault mode accurately. Accumulate the characteristics difference with the sliding window, and then:(27)$${\tilde{r}}_{k}=\sum_{j=k-N_{0}+1}^{k}r_{j},$$
where *N*_0_ is the window length. The characteristics difference ${\tilde{r}}_{k}$ obeys the distribution as follows:(28)$${\tilde{r}}_{k}∼N\left(\sum_{j=k-N_{0}+1}^{k}z_{j},N_{0}\Sigma \right).$$

The ratio $\left|\sum_{j=k-N_{0}+1}^{k}z_{j}/\sqrt{N_{0}\Sigma }\right|$ between the characteristics difference and the noise increases significantly. Therefore, the characteristic difference between two modes can be increased with a sliding window, and the fault identification rate may be increased.

### 4.3. Fault Identification Step Based on Improved Neural Network

According to the discussions in Section 2, due to the adjustment function of a closed-loop control system, the coupling effect among different modes becomes stronger, which brings greater difficulty for fault identification. Thus, in Section 4.2, a fault identification method based on improved deep neural network is proposed. The steps for the proposed fault identification method are given as follows:

Step 1. Preprocess the tagged training data: set an appropriate sliding window size, and then rearrange the data according to the sliding window;

Step 2. Select an appropriate network structure [25]: preset the number of hidden layers, the node number of the input layer, the output layer and each hidden layer to establish a deep neural network structure;

Step 3. Feed the training data into the deep neural network and select an appropriate activation function for the supervised training process;

Step 4. Feed the test data into the trained deep neural network, and record the output result to identify the test data;

Step 5. Calculate the identification rate of the whole test data and each data mode, respectively, and obtain the relationship between the identification rate and the window size.

## 5. Simulations

In this section, three simulations that verify the feasibility of the proposed fault identification approach in closed-loop control systems based on the improved deep neural network are included. The simulation cases include the numerical simulation model, the Tennessee Eastman Process (TEP) based on the industrial process simulation model and the satellite attitude control system (SACS).

### 5.1. Case 1: Numerical Simulation Model

The total amount of simulation data is 30,000 sets according to Equation (2). There are three modes, i.e., a normal mode and two fault modes. The data sample number of each mode is 10,000, and the data dimension is 1.

The first fault mode is process fault and the second is sensor fault, of which the fault amplitude are 0.3 and −0.3, respectively. The parameter matrices are $A=0.5I_{1\times 1}$, $B=0.5I_{1\times 1}$, $C=I_{1\times 1}$, $A^{c}=0.7I_{1\times 1}$, $B^{c}=0.3I_{1\times 1}$, $C^{c}=0.5I_{1\times 1}$, $D^{c}=0.5I_{1\times 1}$. The state noise and the measurement noise are independent of each other, and both obey the normal distribution with zero mean and the standard deviation is 0.1. The initial state $x_{1}=10$, and the constant variables $\tilde{u}=10$, ${\tilde{x}}^{c}=10$, $\tilde{y}=10$. Then the numerical simulation data is obtained, as shown in Figure 3.

Firstly, examine the case that there is no closed control loop, i.e., the control input vector of the system is given artificially. To simulate an open-loop control system, remove the closed control loop, and set the expected control input vector $\tilde{u}$, then the simulation data set is obtained, as shown in Figure 4.

As can be seen from Figure 3 and Figure 4, the difference between the measurement output vectors of the closed-loop system of any two modes is much smaller than that of the open-loop system. The measurement mean of the normal data, the first fault data and the second fault data were 10.001, 10.300, 9.850 in the closed-loop control system, respectively. Meanwhile, they were 9.998, 10.602, 9.702 in the open-loop control system, respectively. When taking the control input vector of the closed-loop control system as the horizontal axis, the measurement output vector as the vertical axis, the simulation data of three modes can be presented in Figure 5.

From Figure 5, the normal data and the second mode data are coupled with each other. If the control input and the measurement output of the current time are directly used as the input nodes of the deep neural network without a sliding window, it will be difficult to identify the fault mode, especially in the coupling parts. Therefore, the sliding window is necessary and used as follows. Three hidden layers are adopted to establish the deep neural network. The total iteration number is 2000. Both the first hidden layer with 10 nodes and the second with five nodes apply Sigmoid function, while the third with three nodes applies the softmax function [25]. We used 70% of data with each mode to train the deep neural network, and the remaining 30% of the data was used as a test dataset. Table 1 lists the relationship between the sliding window size and the identification rate, as well as the training accuracy.

From Table 1, the identification rate of various modes increases and the training accuracy is more accurate with the increase of the window size. It shows that the method using sliding windows to improve deep neural networks is effective.

The identification rates with the traditional methods introduced in Section 3 are also shown in Table 2. Obviously, compared with traditional distance-based or angle-based identification methods, the improved deep neural network fault identification method improves the fault identification rate effectively.

### 5.2. Case 2: TEP Simulation Model

The Tennessee-Eastman Process (TEP) system is based on an industrial process simulation model, which was created by an American company named Eastman in 1993. TEP can provide a realistic and usable industrial process for evaluating the process monitoring and control methods [19,26]. A large amount of literature refers to it as a data source for research on control, optimization, process monitoring, and fault diagnosis and so on. Figure 6 shows the flow chart of TEP industrial equipment.

The TEP system consists of five parts: reactor, condenser, compressor, steam/liquid separator and stripper. The equipment is operated under a closed-loop controller, and a total of 41 variables are collected. In this experiment, three modes were simulated, namely normal data, A/C feed ratio fault, and B-component feed fault. The simulation time was set to 50 h, and the sampling period was 0.01 h. Hence, 5000 sets of data were sampled in total. The variables in the three modes for the reactor are plotted in Figure 7, including reactor feed rate, reactor pressure, reactor grade and reactor temperature, respectively. The data can be gotten from the webpage: http://depts.washington.edu/control/LARRY/LE/download.html.

The first 3500 sets of data in each mode were trained in the deep neural network, and the remaining 1500 sets of data were used for test. The total number of iterations was 100. There were three hidden layers in the deep neural network. The first is with 100 nodes and the second is with 50 nodes. Both layers apply Sigmoid function, while the third with three nodes applies the softmax function. Table 3 shows the relationship between the window size and the identification rate, as well as the training accuracy.

Similarly, from Table 3, the identification rate of various modes increases and the training accuracy is more accurate with the increase of the window size. The identification rates with the traditional methods are shown in Table 4. Obviously, compared with the traditional distance-based or angle-based identification methods, the improved deep neural network fault identification method improves the fault identification rate effectively. It indicates that the proposed method can be applied for fault identification in TEP systems and other industrial systems.

### 5.3. Case 3: SACS Simulation Model

As shown in Table 5, there are seven variables in the satellite attitude control system (SACS). They are earth sensors in the roll ($e_{r}$) and pitch ($e_{p}$) directions, sun sensors in the roll ($s_{r}$) and pitch ($s_{p}$) directions, gyroscopes in the roll ($g_{r}$), pitch ($g_{p}$) and yaw ($g_{y}$) directions. Three modes of data are collected, which are the normal, $e_{p}$ fault and $s_{p}$ fault. The data set is shown as Figure 8.

There were 501 sets of the normal data and 143 sets of both fault data. Half of the each mode data is used to train the deepneural network, and the structure and the activation functions are the same as the Section 5.2’s. It is shown that the identification rate of various modes increases with the increase of the window size in the Table 6. The identification rates with the traditional methods are shown in Table 7. Obviously, compared with the traditional distance-based or angle-based identification methods, the improved deep neural network fault identification method improves the fault identification rate effectively in the SACS.

## 6. Conclusions

In this paper, fault identification for closed-loop control systems has been studied. The characteristics of closed-loop control systems were analyzed, the propagation mechanism of faults inside of the system was researched and their influence on measurement output vectors theoretically. The difficulty of fault identification for closed-loop control systems was revealed. The deep neural network method has been proposed. Then the sliding window technique has been used to improve the deep neural network method, and its feasibility for fault identification for closed-loop control systems has been proved. Finally, numerical simulations, the Tennessee-Eastman system and the satellite attitude control system have been used to verify the feasibility of the fault identification of closed-loop control systems.

## Figures and Tables

**Figure 1 sensors-19-02131-f001:**
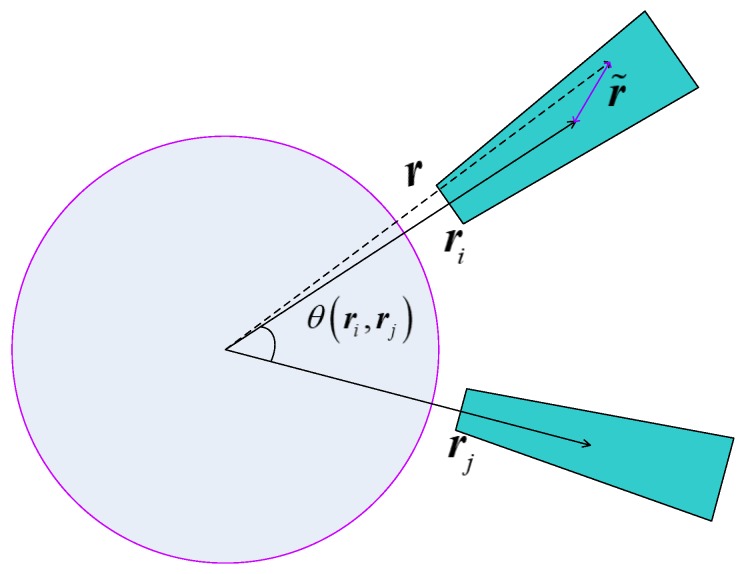
Mode direction and angle identification.

**Figure 2 sensors-19-02131-f002:**
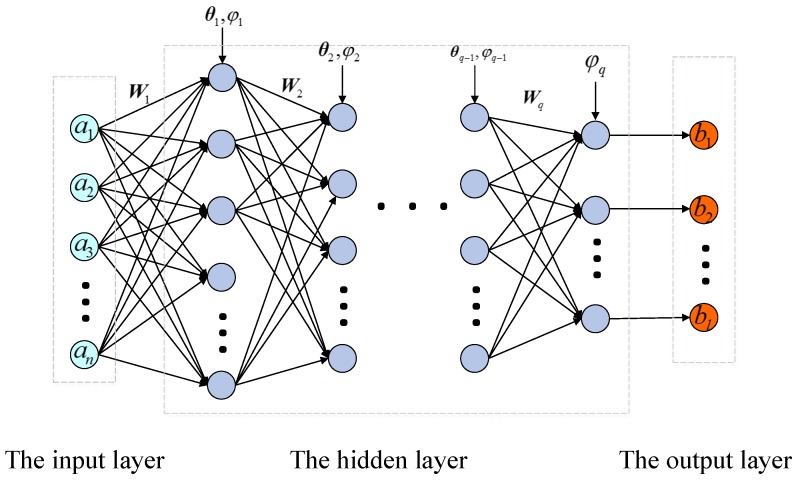
Deep neural network structure.

**Figure 3 sensors-19-02131-f003:**
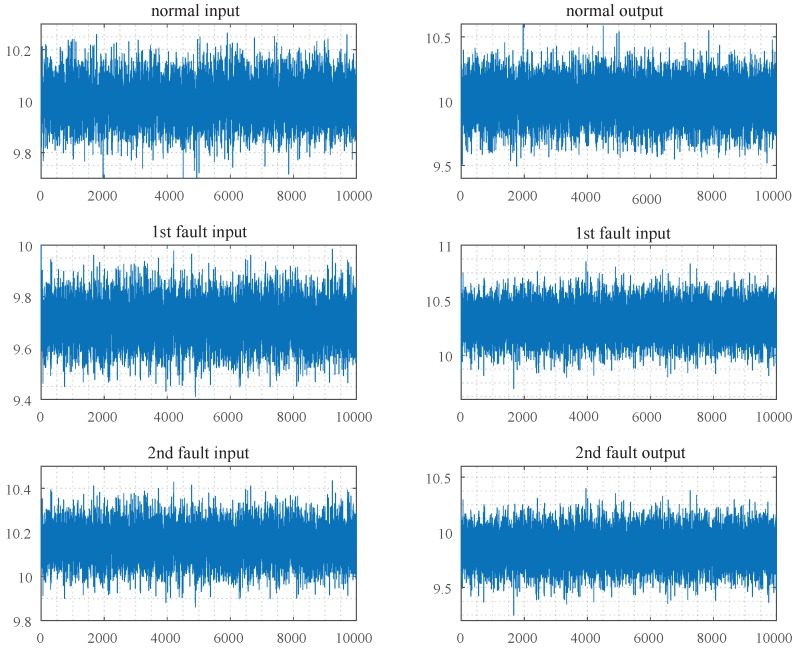
Simulation data set of closed-loop control system.

**Figure 4 sensors-19-02131-f004:**
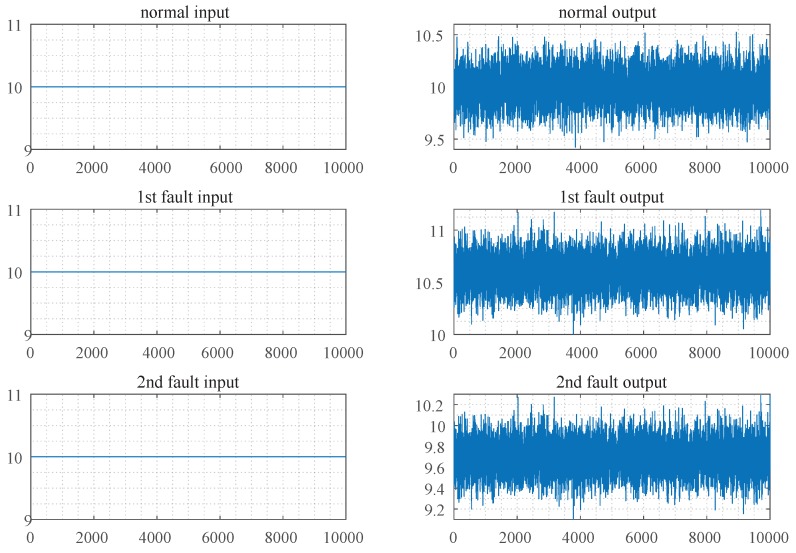
Simulation data set of open-loop control system.

**Figure 5 sensors-19-02131-f005:**
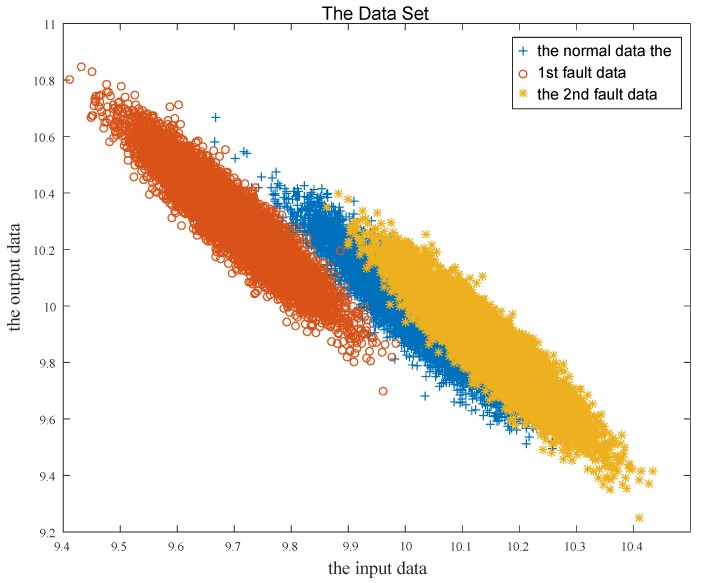
Dataset dot plot.

**Figure 6 sensors-19-02131-f006:**
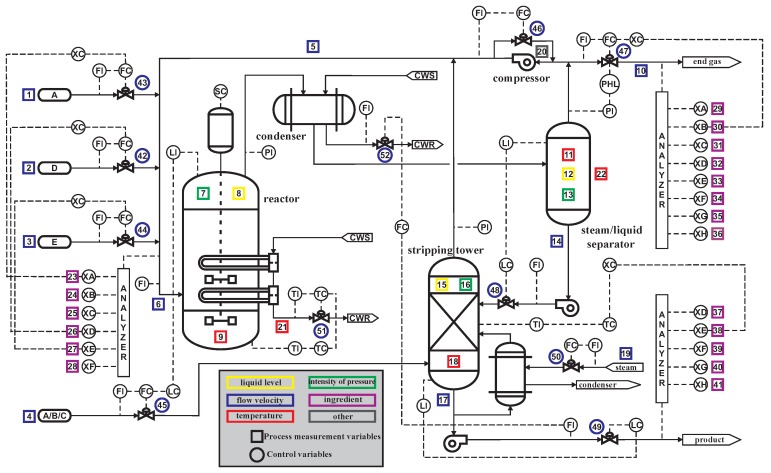
Tennessee-Eastman Process (TEP) industrial equipment flow chart.

**Figure 7 sensors-19-02131-f007:**
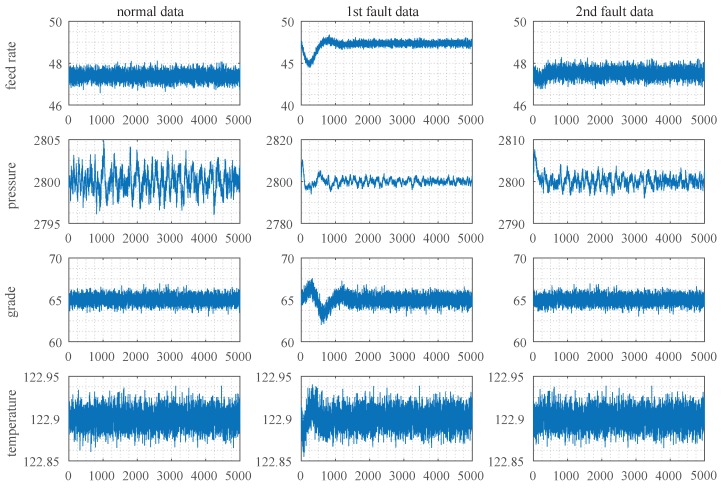
Partial data set.

**Figure 8 sensors-19-02131-f008:**
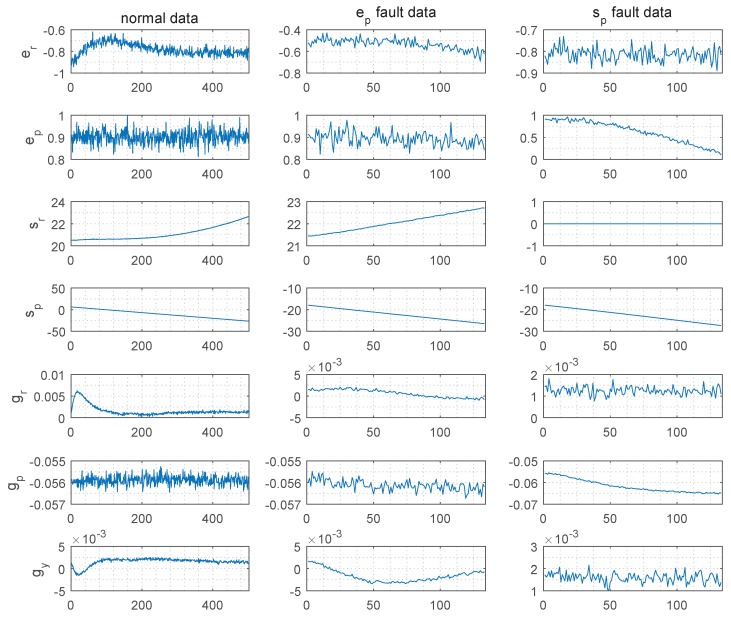
Data set.

**Table 1 sensors-19-02131-t001:** The relationship between the sliding window size and fault identification indicators.

Window Size	Identification Rate	The Normal Data	The First Fault Data	The Second Fault Data	The Accuracy
No window	94.622%	91.467%	99.600%	92.800%	0.0250
2	96.066%	93.462%	99.733%	95.000%	0.0186
3	97.044%	94.963%	99.867%	96.300%	0.0141
4	97.766%	95.896%	99.933%	97.467%	0.0106
5	98.366%	96.796%	99.933%	98.367%	0.00788
6	98.733%	97.095%	99.933%	99.167%	0.00589

**Table 2 sensors-19-02131-t002:** The identification rates of various methods.

Fault Identification Method	Identification Rate	The Normal Data	The First Fault Data	The Second Fault Data
The improved DNN	98.733%	97.095%	99.933%	99.167%
The distance-based method	77.80%	66.03%	91.50%	75.87%
The angle-based method	77.01%	66.47%	90.27%	74.30%

**Table 3 sensors-19-02131-t003:** The relationship between the sliding window length and fault identification indicators.

Window Size	Identification Rate	The Normal Data	The First Fault Data	The Second Fault Data	The Accuracy
No window	99.756%	99.533%	99.867%	99.867%	0.00562
2	99.822%	99.733%	99.933%	99.800%	0.00196
3	99.911%	99.933%	99.867%	99.933%	0.00167

**Table 4 sensors-19-02131-t004:** The identification rates of various methods.

Fault Identification Method	Identification Rate	The Normal Data	The First Fault Data	The Second Fault Data
The improved DNN	99.911%	99.933%	99.867%	99.933%
The distance-based method	94.76%	95.47%	93.07%	95.73%
The angle-based method	62.98%	48.33%	47.67%	92.93%

**Table 5 sensors-19-02131-t005:** The identification rates of various methods.

Variable	Sensor	Variable	Sensor
$e_{r}$	earth sensor in the roll direction	$g_{r}$	gyroscope in the roll direction
$e_{p}$	earth sensor in the pitch direction	$g_{p}$	gyroscope in the pitch direction
$s_{r}$	sun sensor in the roll direction	$g_{y}$	gyroscope in the yaw direction
$s_{p}$	sun sensor in the pitch direction		

**Table 6 sensors-19-02131-t006:** The relationship between the sliding window length and fault identification indicators.

Window Size	Identification Rate	The Normal Data	$e_{p}$ Fault Data	$s_{p}$ Fault Data
No window	96.610%	99.401%	82.707%	100.00%
2	99.608%	100.00%	98.496%	99.248%
3	99.739%	100.00%	98.496%	100.00%

**Table 7 sensors-19-02131-t007:** The identification rates of various methods.

Fault Identification Method	Identification Rate	The Normal Data	$e_{p}$ Fault Data	$s_{p}$ Fault Data
The improved DNN	99.739%	100.00%	98.496%	100.00%
The distance-based method	83.31%	77.25%	89.47%	100.00%
The angle-based method	76.79%	64.47%	100.00%	100.00%

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
