# Peer review of "Fault Identification for a Closed-Loop Control System Based on an Improved Deep Neural Network"

_sensors, 2019, doi:10.3390/s19092131_

Round 1

Reviewer 1 Report

This paper proposes to amplify the fault-to-noise ratio and the characteristics difference of a system through applying a sliding window to the traditional deep neuronal network training. The paper is interesting and sounds scientific for industrial applications, it is well organized, all the equations and variables descriptions are clear.

It is recommended to enrich the introduction with more examples of machine learning research for control feedback, such as the below literature where Outliers elimination and Support Vector Machine (SVM) were used to feedback and correct measurement issues.

Multivariate outlier mining and regression feedback for 3D measurement improvement in opto-mechanical system. Optical and Quantum Electronics48(8), 403.

Combined application of power spectrum centroid and support vector machines for measurement improvement in optical scanning systems. Signal Processing, 2014, vol. 98, p. 37-51.

Regarding the paper contribution:

It is clear that the contribution of this paper is the deep neural network fault identification method based on a sliding window. However, it was not stated if it is a new method or a method variation (no other literature was cited regarding deep neural network training variations).

It would have been valuable also to state the advantages found when were used sliding windows for different algorithms before.

Author Response

The authors thank the reviewer for the encouragement for this article. As for the improvements, we have made the following changes:

In the introduction, we enrich it with more examples of machine learning research for control feedback as the reviewer’s comment. The details are as follows:

For the structural health monitoring in opto-mechanical system, the system scanning technology is based on a mechanical rotary mirror and the computational processing of an optoelectronic signal is applied[1]. The 3D measurement is improved with the multivariate outlier mining and regression feedback. New research performed in the signal and data processing of the 3D measurement system demonstrated the effectiveness of signal processing strategies as a tool for a feedback loop in any measurement system.

For position detection in real life application for Structural Health Monitoring (SHM) by a novel method, Support Vector Machine (SVM) Regression was applied to predict measurements errors for Accuracy Enhancement in Optical Scanning Systems[2]. The accuracy can be enhanced based on the Power Spectrum Centroid Calculation. The SVM Regression Method is proved that it can be used to increase measurement accuracy for Optical Scanning System.

It is an improved deep neural network method in this manuscript, and in the introduction, we add some references about the deep neural network training variations.

[1]    Flores-Fuentes W , Sergiyenko O , Gonzalez-Navarro, Félix F, et al. Multivariate outlier mining and regression feedback for 3D measurement improvement in opto-mechanical system[J]. Optical and Quantum Electronics, 2016, 48(8):403.

[2]    Flores-Fuentes W , Rivas-Lopez M , Sergiyenko O , et al. Combined application of Power Spectrum Centroid and Support Vector Machines for measurement improvement in Optical Scanning Systems[J]. Signal Processing, 2014, 98:37-51.

Reviewer 2 Report

Review 1

The article concerns a very interesting topic of fault diagnostics in closed-loop control systems. The paper has a strong theoretical background and is well structured. However, there have been pointed out some general and detailed remarks, which could improve the quality of the paper.

General remarks:

1.       Authors should visualize the deep neural networks that have been used in practical examples;

2.       In my opinion, there should be presented maybe one or two more examples of faults identified by the presented technique, especially from TEP simulation model,  that would prove the applicability of the shown method.

Detailed remarks:

Line(s):

14: I suggest that Authors remove all ‘we’ in the manuscript;

45-47: If the paper focuses on closed-loop fault identification, it is rather unnecessary to review the research from the area of open-loop fault identification (references 10-14). However, if Authors would like to leave the part as is, I suggest to provide detailed comment connected with the topic of the paper;

336: All references have unnecessary [J]. Therefore Authors should adjust formatting to the journal requirements;

Author Response

The authors thank for the detailed and constructive comments. And the advice is quite beneficial for the paper. The changes are made as follows:

1. In the simulation, we introduce the deep neural network structure and the activation functions according to the Fig. 2 for visualization in practical examples. For the result, it is difficult to visualize because the result is the probability of correct identification. So we show it with the statistical data in a table.

2. We add another simulation based on satellite attitude control system and get a good result. It can prove that this method proposed from this manuscript is of applicability based on the TEP simulation model and the satellite attitude control system.

Detailed changes:

1.     The whole manuscript is examined and all “we” in it are removed.

2.     In the introduction, we remove the references 10-14 and underline the connection between the open-loop control system and the topic of the manuscript.

3.     We examine all references and the format of references is standardized.

Reviewer 3 Report

This paper proposed a deep learning-based approach to identify the sensor faults of control systems. This paper is overall nicely written with clear motivation, solid theoretical analysis, and convincing experimental results. It can be, however, further improved from the following perspectives:

(1) The introduction can be more focused on sensors. Currently, the paper seems to overemphasize on control systems. I understand control systems are the target applications of sensors in this work; however, authors might want to consider how the proposed work could (potentially) shed insights to the research community of sensors. That is to say, what methodological contributions can this work make to sensor researchers and practitioners? 

(2) There are tons of equations in the proposed approach without much high-level intuition. I think for the main body of the research paper, it's more important to explain a problem/solution in a plain language and the long mathematical derivation can be moved to the appendix. 

(3) For the deep learning section, I think some content can be moved to the background section because they are very basics. For instance, Figure 3 is more like a diagram from a deep learning 101 course and might not look appealing to the readers.

(4) For evaluations, authors might want to elaborate a bit more. There are a few issues here: (a) Please provide the references to the tested cases, e.g., Tennessee Eastman Process; (b) What type of hardware and software did those experiments run on? (c) I'm not sure how the deep-learning approach is implemented. Is it implemented from scratch or do the authors take some open-source deep learning frameworks like TensorFlow? If in the latter case, how did the authors integrate the framework into the simulator?

Author Response

The authors are grateful for the advice for this manuscript. The specific explanation is given as follows:

(1)   In this manuscript, the measurement data is from the sensors, so this manuscript is connected with sensors closely. This manuscript makes contributions for the problem how to process the data from sensors. And we enrich the introduction with the sensors.

(2)   There are two parts, subsection 2.2 and subsection 2.3, in which the equations are too many. For this question, we reserve the first part to explain the complete process of the sensor fault propagation mechanism, and we move the second part to the appendix.

(3)   Since the deep neural network leads to the improved method, this part is simplified but reserved. The basic statement is omitted, for example, the process model of deep neural network. But it is difficult to introduce the training regulation and the activation function without the deep neural network structure.

(4)   We perfect the simulation and the specific explanations are as follows:

a)      The Tennessee Eastman Process’ data is got from the webpage: http://depts.washington.edu/control/LARRY/LE/ download.html.

b)      It is an industrial process simulation model and it can provide a realistic and usable industrial process for evaluating the process monitoring and control by simulation. The process is simulated by computer and the TEP system is usually used to industrial fault diagnosis.

c)      The simulation is done by matlab and there is neural network toolbox online. The code can be downloaded from the webpage: https://download.csdn.net/download/weixin_43969938/10890236. And we did some modification on the code to satisfy the needs in the manuscript.
